# Antioxidant Activities of Alkyl Substituted Pyrazine Derivatives of Chalcones—In Vitro and In Silico Study

**DOI:** 10.3390/antiox8040090

**Published:** 2019-04-05

**Authors:** Višnja Stepanić, Mario Matijašić, Tea Horvat, Donatella Verbanac, Marta Kučerová-Chlupáčová, Luciano Saso, Neven Žarković

**Affiliations:** 1Division of Molecular Medicine, Ruđer Bošković Institute, Bijenička cesta 54, HR-10000 Zagreb, Croatia; Tea.Horvat@irb.hr; 2Department for Intercellular Communication, Centre for Translational and Clinical Research and Croatian Centre of Excellence for Reproductive and Regenerative Medicine, School of Medicine, University of Zagreb, Šalata 2, HR-10000 Zagreb, Croatia; mario.matijasic@mef.hr (M.M.); dverbanac@pharma.hr (D.V.); 3Department of Medical Biochemistry and Haematology, Faculty of Pharmacy and Biochemistry, University of Zagreb, Domagojeva 2, HR-10000 Zagreb, Croatia; 4Department of Pharmaceutical Chemistry and Pharmaceutical Analysis, Faculty of Pharmacy, Charles University, Heyrovského 1203, 500 05 Hradec Králové, Czech Republic; kucerom@faf.cuni.cz; 5Department of Physiology and Pharmacology Vittorio Erspamer, Sapienza University, P.le Aldo Moro 5, 00185 Rome, Italy; luciano.saso@uniroma1.it

**Keywords:** antioxidant, chalcone-like, DFT, in silico, in vitro, pyrazine, radical scavenging

## Abstract

Chalcones are polyphenolic secondary metabolites of plants, many of which have antioxidant activity. Herein, a set of 26 synthetic chalcone derivatives with alkyl substituted pyrazine heterocycle A and four types of the monophenolic ring B, were evaluated for the potential radical scavenging and antioxidant cellular capacity influencing the growth of cells exposed to H_2_O_2_. Before that, compounds were screened for cytotoxicity on THP-1 and HepG2 cell lines. Most of them were not cytotoxic in an overnight MTS assay. However, three of them, 4a, 4c and 4e showed 1,1-diphenyl-2-picrylhydrazyl (DPPH●) radical scavenging activity, through single electron transfer followed by a proton transfer (SET-PT) mechanism as revealed by density functional theory (DFT) modeling. DFT modeling of radical scavenging mechanisms was done at the SMD//(U)M052X/6-311++G** level. The in vitro effects of 4a, 4c and 4e on the growth of THP-1 cells during four days pre- or post-treatment with H_2_O_2_ were examined daily with the trypan blue exclusion assay. Their various cellular effects reflect differences in their radical scavenging capacity and molecular lipophilicity (clogP) and depend upon the cellular redox status. The applied simple in vitro-in silico screening cascade enables fast identification and initial characterization of potent radical scavengers.

## 1. Introduction

Chalcones are molecules containing a 1,3-diphenylprop-2-en-1-one fragment (Figure 1). They are natural products from the group of flavonoids. Plenty of chalcones have been isolated from natural plant sources [1]. However, numerous synthetic chalcone derivatives with various substitutions at the benzene rings or with heterocyclic analogs have also been prepared in laboratories [2]. From the synthetic aspect, the advantage of chalcones is the structural versatility generated by relatively simple synthetic procedures [2,3,4].

Herein, we have considered chalcone-like compounds with non-substituted or 4′-alkyl substituted pyrazine heterocycle at the place of benzene ring A (Figure 1, Table 1). Some biological activities of these synthetic derivatives have already been described. Chalcones and their various heterocyclic analogues exert modulatory effects on different molecular targets depending on the aromatic system types and the substitution pattern [4,5]. The pyrazine analogues of chalcones have been screened for antimicrobial and antifungal activity, but the derivatives considered here have shown no significant anti-infective effects [6,7,8]. Antiproliferative activity of chalcones has also been tested on various cancer cell lines [9]. Chalcones exert cytotoxic effects on various cancer cell lines and their use as antitumor agents with various molecular mechanisms of action, has been discussed in several recently published reviews [5,10,11]. However, many chalcones with significant pharmacological effects are not toxic to non-tumorous cells and for that reason chalcones are suggested for cancer chemoprevention [1,12]. Cancer chemopreventative activity has been demonstrated for natural chalcones, e.g., xanthohumol (*Humulus lupulus*, Cannabaceae), naringenin chalcone and its glucoside isosalipurpuroside (*Helichrysum maracandicum*, Asteraceae), isobavachalcone (*Angelica keiskei*, Apiaceae), as well as for various synthetic hydroxylated and methoxylated chalcones (Figure 1) [13,14,15]. Chemopreventative effects of chalcones have been recognized in respect to their interference with cellular mechanisms involved in mutagenesis and/or the repair of damaged DNA, as well as with chalcone related increases in phase 2 metabolic enzyme activity [16].

Biological effects of chalcones have also been ascribed to their antioxidative activities [18]. Chalcones exert indirect and direct redox activities. They are inhibitors of aldose reductase ALR2, an enzyme with antioxidative and anti-inflammatory effects [19,20,21]. There are also chalcones with direct radical scavenging (RS) activities [18,22]. The direct RS activity of natural chalcones such as prenylated xanthohumol [15], non-prenylated butein [15,23] or various licorice chalcones is well-documented [24]. The presence of free hydroxyl groups (at C-2′ in the ring A and a catechol group in the ring B) and the α,β-double bond in the linker are considered important structural features for efficient antiradical activity of chalcones [15].

The aim of our study was to explore the antioxidant potential associated with the RS mechanism for the set of synthetic pyrazine analogues of chalcones [19,25]. Their antioxidant activities were measured in vitro through direct reaction with a 1,1-diphenyl-2-picrylhydrazyl radical (DPPH•) and additionally in the THP-1 cellular assay treated with hydrogen peroxide (H_2_O_2_). The RS mechanism was characterized by performing quantum-chemical density functional theory (DFT) modeling. The observed difference in antioxidant cellular activity of good radical scavengers was ascribed to differences in their RS capacity and molecular lipophilicity.

## 2. Materials and Methods

### 2.1. Studied Compounds

All studied chalcone and chalcone-like compounds were synthesized previously according to the cited references [6,7,8,19]. The structural characteristics of all compounds (melting points, IR and NMR data) are available: for compounds 1, 2, 3 and 4 in the supplementary material of Reference [19], 1a, 1b, 1d, 1e, 1f, 3a, 3b, 3d, 3e, 3f, 4a, 4e and 4f in Reference [6], 2a, 2b, 2d, 2e and 2f in Reference [7] and 1c, 2c, 3c and 4c in Reference [8]. The compounds were dissolved in dimethyl sulfoxide (DMSO).

### 2.2. Cell Lines

Adherent liver hepatocellular carcinoma cell line HepG2 (ATCC, HB-8065) and human acute monocytic leukemia cell line in suspension THP-1 (ATCC, TIB-202) were purchased from American Type Culture Collection (ATCC, Manassas, VA, USA). Cell lines were maintained in complete DMEM/F12 medium (Sigma, D8437, St. Louis, MO, USA) and complete RPMI1640 (Sigma, R7388, St. Louis, MO, USA) respectively, supplemented with 10% Fetal Calf Serum (FCS, Sigma, F7524, St. Louis, MO, USA) at 37 °C in 5% CO_2_ atmosphere.

### 2.3. Cytotoxic Activity

A cytotoxicity assay was performed using MTS CellTiter 96 AQueous One Solution Cell Proliferation Assay (Promega, G3580, Madison, WI, USA) [26]. Double dilutions of compounds were prepared in the concentration range of 100–0.2 µM within microplate wells. In each well, 5 × 10^4^ cells were added and the plates were incubated overnight at 37 °C in 5% CO_2_ atmosphere. Following this, 10 µL of MTS (3-(4,5-dimethylthiazol-2-yl)-5-(3-carboxymethoxyphenyl)-2-(4-sulfophenyl)-2H -tetrazolium) reagent was dispensed per well and the plates were incubated for 1–6 h at 37 °C in 5% CO_2_ atmosphere. Control wells consisted of media only (blank) or cells with 1% DMSO added (control). The absorbance was recorded at 490 nm using a Wallac Victor2 microplate reader (PerkinElmer, Waltham, MA, USA). Results were analyzed in GraphPad Prism software (GraphPad Software, San Diego, CA, USA).

### 2.4. DPPH• Radical Scavenging Assay

The DPPH (1,1-diphenyl-2-picrylhydrazyl, Sigma, D9132, St. Louis, MO, USA) method was used to determine the RS activity of compounds [27]. Vitamin C (Acros Organics, Geel, Belgium) was used as a standard antioxidant control and was prepared in the same way as tested compounds. Dilutions of tested compounds and vitamin C were prepared in a final concentration range of 1 mM–0.01 μM. One milliliter of compound solution was added to 1 mL of freshly prepared DPPH• solution (3.9 mg/50 mL ethanol; final concentration of 100 µM) and the reaction mixture was incubated in the dark at room temperature for 30 min. Absorbance was recorded at 540 nm using a 96-well Wallac Victor2 plate reader. Neutralization of DPPH• radical by a compound was calculated according to the following formula:RS activity (%) = [(A_0_ − A_1_)/A_0_] × 100(1)
where A_0_/A_1_ is the control/sample absorbance. The results were averages of three measurements. The IC_50_ value was calculated using GraphPad Prism software (GraphPad Software, San Diego, CA, USA).

### 2.5. Effects of Compounds on the Growth of THP-1 Cells Treated with H_2_O_2_

The substances without cytotoxic effects and with RS activity were tested for antioxidant potential at the cellular level using THP-1 cells treated by H_2_O_2_ [28]. The cells were exposed to the IC_50_ concentrations of the compounds determined by the DPPH• assay. Compounds were applied 30 min either before or after exposure to 5 mM of H_2_O_2_, which is equivalent to the cytotoxic ED_50_ dose of H_2_O_2_ for the THP-1 cells [28]. The cell growth was analyzed every 24 h, via the trypan blue exclusion assay using a Bürker-Türk hemocytometer (Brand GMBH + CO KG, Wertheim, Germany), during a 4-day period. All assays were carried out in triplicates. The comparison of mean values was done by the two-tailed Student’s *t*-test.

### 2.6. In Silico Methods

#### 2.6.1. Predictions of Physicochemical and ADME/Tox Properties

Physicochemical and ADME/Tox (Absorption, Distribution, Metabolism and Excretion / Toxicity) parameters were predicted by the algorithms DataWarrior [17] and SwissADME [29]. Inputs for both algorithms were SMILES (simplified molecular-input line-entry system) (Appendix A).

#### 2.6.2. Molecular Similarity

The MACCS fingerprint generation and molecular similarity clustering using binary distance were done by R packages *rcdk* and *factoextra*, respectively [30,31].

#### 2.6.3. DFT Calculations

The gas-phase (g) reaction parameters: bond dissociation enthalpy (BDE), ionization potential (IP), acidity and electron transfer enthalpy (ETE), as well as their aqueous (aq) free energy (FE) counterparts BDFE_aq_, IFE_aq_, pK_a_, and ETFE_aq_, respectively, were calculated by using thermochemical cycles [32] along with the DFT model (U)M052X/6-311++G** implemented in Gaussian 09 [33,34]. Equilibrium geometries in neutral and anionic closed-shell as well as neutral mono-radical and radical cation open-shell doublet ground electronic states, were optimized in the gas phase. The minima were confirmed by no imaginary vibrational frequencies. Calculations were done with the compounds in *E*-configuration [6,7,8,19]. The free energies of hydration Δ*G**_hyd_ were determined at the gas phase geometries by using the universal continuum solvation model SMD [35].

## 3. Results and Discussion

### 3.1. In Vitro Cytotoxicity

The cytotoxic effects of 26 compounds were evaluated by using the MTS test which has been run as a part of routine safety compound profiling [26,36]. The compounds (applied in concentrations up to 100 μM overnight) showed no considerable effects on cellular metabolism and viability of HepG2 and THP-1 cells (Table 1). None of compounds were cytotoxic against HepG2 cells. In regard to THP-1 cells, stronger cytotoxicity was detected only for the two compounds: the 2-OH substituted derivate 1d and the 3-OCH_3_, 4-OH substituted derivate 4f, for which the IC_50_ values were 27 µM and 36 µM, respectively. In addition, compounds with single 2- or 3-OH group in the ring B showed, in average, somewhat stronger cytotoxicity as compared with the 4-OH analogues (Table 1) [37].

However, some of the studied compounds have been determined to exert cytotoxic effects during a longer incubation period (Figure 2) and/or against another cancer cell lines. For example, the chalcone **3** had significant cytotoxicity against cell lines A549, PC3, MCF-7, HT-29 and WRL68 [9]. Many chalcone analogues with heteroaryl A or B ring(s) (e.g., indole or quinolone analogues) have also been reported to exhibit potent growth inhibitory activity on cancer cell lines [5,10].

### 3.2. In Vitro and in Silico Radical Scavenging Analysis

The various types of redox activity of the studied pyrazine chalcone derivatives have been demonstrated so far [6,19]. Some of them inhibited the catalytic activity of aldose reductase ALR2. The pyrazine derivatives 2a and 3a showed moderate ALR2 inhibition activity. Regarding the corresponding chalcones, only 3 had comparable ALR2 inhibition. Neither chalcones nor their pyrazine analogues with 2-OH or 3-OCH_3_, 4-OH substitutions showed this activity. The 2-OH (series 1) and 4-OH (series 3) substituted compounds with a 4′-alkylated pyrazine ring A (Table 1) were also found to impair the photosynthetic electron transport system in plants in the study of their herbicide activity, which was conducted since the pyrazine chalcone derivatives have common structural features with the cinnamoyl pyrazine plant growth inhibitors [6].

Chalcones have also been tested in various RS assays [19,38]. When tested in the assay of styrene auto-oxidation under oxygen atmosphere at 50 °C initiated by azo-bis-isobutyronitrile, 1a, 3a and 4a showed significant RS activity comparable to or even greater than that of the corresponding *para*-coumaric acid fragment [38]. Antioxidative activity of chalcones 3 and 4 and their pyrazine analogues 3a and 4a, respectively, have been measured in the DPPH• assay and the compounds with the guaiacyl-like ring B (series 4) were determined to have moderate RS capacity [19]. In our study, the DPPH• RS activity was measured for other synthesized derivatives as well (Table 1).

Among all tested compounds, only the guaiacyl-substituted derivatives 4a, 4c and 4e have displayed DPPH• RS potential with IC_50_ values of 186 µM, 39 µM and 46 µM, respectively (Table 1). They were weaker DPPH• radical scavengers than the reference vitamin C which has an IC_50_ value of 15 µM. It is well-known that phenolic compounds with high RS activity generally have a catechol fragment [32,39,40,41]. The elimination of one of the catecholic OH groups e.g., by methylation, diminishes the RS capacity of polyphenols [40].

The obtained DPPH• RS data were interpreted and the underlying RS mechanism of the guaiacyl-substituted derivatives was determined by applying quantum-chemical modeling (Table 2). Three mechanisms have been proposed for RS by phenolic compounds [32]. It has been proposed that neutral phenolic compounds may scavenge free radicals by donating an H-atom in a one-step hydrogen-atom transfer (HAT) mechanism which is described by the O-H BDE parameter. In a simple model designed for the estimation of relative RS capacities, the parameter BDE is defined as a difference between a sum of enthalpies of the phenoxy radical and hydrogen atom products and an enthalpy of a neutral phenolic reactant [32]. Another proposed RS mechanism for neutral phenols is a two-step process composed of a single electron transfer followed by fast proton transfer from the positively charged radical intermediate (SET-PT). The first step is described by the ionization potential (IP) of a phenolic molecule. In the third proposed RS mechanism, the two-step sequential proton loss electron transfer (SPLET), following the first step of deprotonation of a phenolic group (described by its acidity constant), an intermediate anion donates an electron with capacity described by the parameter ET(F)E [32]. Assuming that 4′-alkyl substituents do not considerably influence the RS properties of the free -OH group in the ring B [41], the parameters were calculated for the compounds with an unsubstituted ring A (a-compounds) and their chalcone analogs (Table 2).

By comparing the parameter values with those of the good radical scavengers, quercetin and vitamin C, only compounds with the guaiacyl group (series 4) are predicted to have moderate RS activity. This has been in agreement with the in vitro results (Table 1). Since the prediction is based on the comparison of the IP_g_ and IFE_aq_ values, the associated RS mechanism of these derivatives has been predicted to be SET-PT [32]. Although the ETFE_aq_ values of most of the pyrazine derivatives are comparable with those of reference compounds (Table 2), since they are weak acids (pK_a_ > 10.0 except for 1a, Table 2), SPLET is not expected to be their dominant RS mechanisms. If it were an important RS mechanism, then the compounds with a 3-OH group would have also shown DPPH• RS activity.

### 3.3. Effects of Selected Compounds on the Growth of THP-1 Cells Pre- or Post-Treated with H_2_O_2_

The three compounds 4a, 4c and 4e which were non-toxic for THP-1 cells after overnight incubation and had demonstrated RS activity, were further tested for the antioxidative activity using THP-1 based cellular assay (Figure 2). They were added at their DPPH• RS IC_50_ concentrations (Table 1) to THP-1 cells 30 min before or after H_2_O_2_. The aim of this part of the study was to see if the compounds may alter cytotoxic and growth suppressing effects of H_2_O_2_ as was recently described for antioxidative 1,4-dihydropyridine derivatives [42]. Although the three compounds are structurally similar (Figure 3), their effects on the growth of THP-1 cells through a four day period without or with the presence of H_2_O_2_ oxidant, were different.

Considering antiproliferative effects of the pure substances, only 4e had no considerable effects on the growth of the THP-1 cells during a longer time period (Figure 2a). In the experiments with H_2_O_2_, all three compounds attenuated pro-oxidative activity of H_2_O_2_, except 4a when it was added to the cell culture after H_2_O_2_ (Figure 2b). The attenuating effects followed a similar pattern, but with a difference in magnitude. The lowest effect against cytotoxic activity of H_2_O_2_ was observed with 4a, which is the weakest radical scavenger and the least lipophilic molecule among them (Table 1). The most efficient reduction of cytotoxic effect of H_2_O_2_ was achieved with 4c when it was added before H_2_O_2_ (Figure 2c), possibly due to its strongest RS capacity. In this treatment, the number of the THP-1 cells did not decrease at all and remained higher than the respective values for the cells treated only with H_2_O_2_ during whole experiment. THP-1 cells in all cases (b)–(d) start to recover after initial damage at 48 h point, but the extent of their recovery was considerably greater in the case of cells treated with the compounds 4c and 4e.

Stronger antioxidative and cellular growth recovery effects of 4c and 4e as compared to 4a, may be associated with their higher RS capacity and additionally with their greater lipophilic character which may enable more extensive uptake of these compounds by THP-1 cells (Table 1, Figure 3) [43]. The alkyl isopropyl and isobutyl R1 substituents significantly increase molecular lipophilicity of 4c and 4e as compared with unsubstituted analog 4a (Table 1).

The three guaiacyl-like derivatives as well as all other studied derivatives satisfy the Lipinski rule of five (Appendix A). They are neutral molecules at pH 7.4 (except 1a, Table 2), moderately soluble in water and of moderate lipophilicity (Appendix A). They are also not predicted to be substrates of the drug efflux pump P-glycoprotein. Thus, these alkyl-substituted pyrazine analogs of chalcones are expected to pass through cellular membranes including those of intestinal epithelial cells. None of the studied compounds are expected to cause mutagenic or tumorigenic effects (Appendix A). However, as α,β-unsaturated carbonyls, these derivatives may also react as Michael acceptors with reactive sulfhydryl groups of glutathione or cysteine amino acid residues of proteins. Such modifications of biomolecules are associated with an induction of the transcription of phase 2 enzymes and elevation of glutathione. These are major protective cellular responses against toxic and carcinogenic substances and may also contribute to the observed antioxidant cellular effects of considered compounds [16,37]. The direct RS activity that the compounds contribute to the observed cellular effects is indicated through the greater attenuation of H_2_O_2_ effects by the stronger radical scavengers 4c and 4e in the period up to 48 h (Figure 2). However, the greater recovery effects after 48 h in the case of THP-1 cells treated with these two compounds may suggest their bifunctional antioxidant role. After a long treatment, 4c and 4e may exert their antioxidant activity through the activation of the Kelch-like ECH-associated protein 1 (Keap1)/ Nuclear factor erythroid 2-related factor 2 (Nrf2) pathway. Some chalcones and chalcone-like derivatives have already been found to activate the Keap1/Nrf2 antioxidant response element (ARE) pathway through a Michael addition to the cysteine residues of Keap1, which is a redox sensor and negative regulator of Nrf2 [44,45].

## 4. Conclusions

The chalcone skeleton is a privileged scaffold in medicinal chemistry [1,2]. Many chalcone derivatives have shown biological activities (anti-infective, anticancer, anti-inflammatory) that may be associated with their antioxidant efficiency [11]. By applying the efficient and relatively inexpensive in vitro–in silico screening cascade, we determined antioxidant profiles for the set of non-cytotoxic pyrazine analogs of chalcones and analyzed them in terms of calculated RS parameters and molecular properties. Only derivatives 4a, 4c and 4e with a monomethylated catechol group in the ring B had DPPH• RS activity (Table 1) [15]. According to the quantum-chemical modeling, their RS mechanism corresponds to the two-step single electron transfer followed by proton transfer (SET-PT) pathway (Table 2). At the cellular level, the extent of attenuating effects against pro-oxidant H_2_O_2_, observed for radical scavengers 4a, 4c and 4e (Figure 2) were in agreement with their relative RS capacities and molecular lipophilicity (clogP) and depend also on the redox status of the cell. The screening cascade combining in vitro assays with proper in silico methods provided the efficient platform for identification and initial characterization of antioxidant properties of the studied compounds.

## Figures and Tables

**Figure 1 antioxidants-08-00090-f001:**
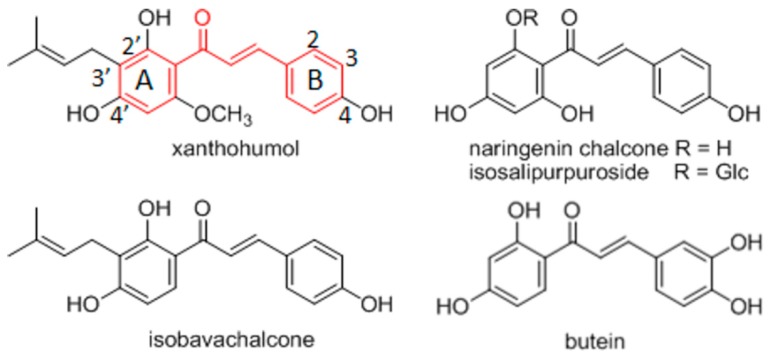
Natural chalcone antioxidants [1]. The 1,3-diphenylprop-2-en-1-one fragment is marked red and the atom numbering used for all studied compounds is labelled.

**Figure 2 antioxidants-08-00090-f002:**
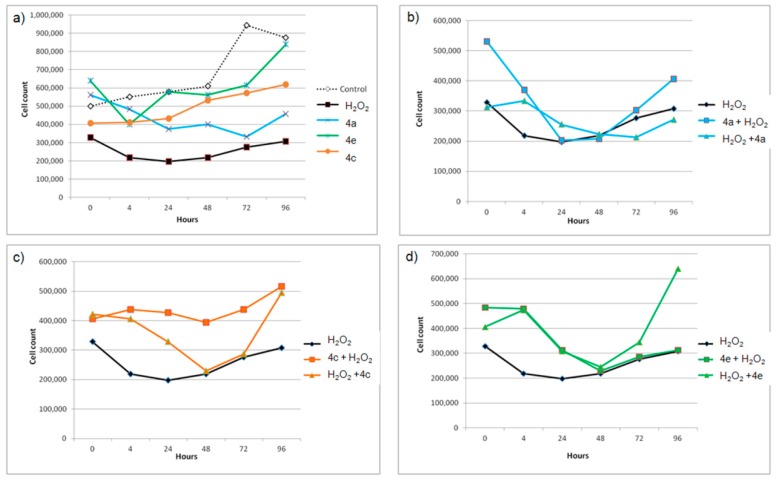
(**a**) The effects of the radical scavengers 4a, 4c and 4e and the oxidant H_2_O_2_ on the growth of THP-1 cells cultured for 96 h; (**b**–**d**) the effects of the compounds given 30 min either before (compound + H_2_O_2_) or after (H_2_O_2_ + compound) H_2_O_2_ on the THP-1 cells.

**Figure 3 antioxidants-08-00090-f003:**
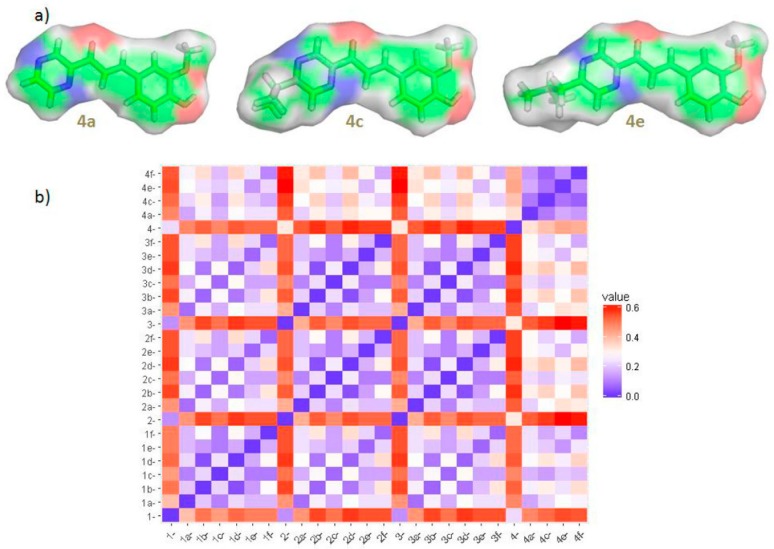
(**a**) Equilibrium ground-state structures (M052X/6-311++G**) with a Connolly surface (white parts correspond to nonpolar surface area) of the radical scavengers 4a, 4c and 4e; (**b**) Clustering of 26 compounds (Table 1) according to their structural similarity (more blue/red—more similar/dissimilar compounds).

**Table 1 antioxidants-08-00090-t001:** The pyrazine analogues of chalcones tested for cytotoxicity against cell lines HepG2 and THP-1 (IC_50_ (μM)) and on DPPH● radical scavenging (RS) activity (IC_50_ (μM)). Their lipophilicity was estimated in silico.

Compound	X	R1	R2	HepG2	THP-1	DPPH● ^1^	clogP ^2^
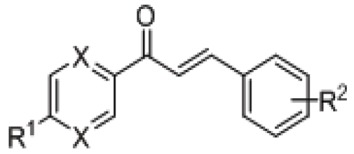							
1	CH	H	2-OH	>100	70	>1000	2.96
1a	N	H	2-OH	>100	40	>1000	1.01
1b	N	propyl	2-OH	>100	>100		2.28
1c	N	isopropyl	2-OH	>100	96	>1000	2.25
1d	N	butyl	2-OH	86	27	>1000	2.73
1e	N	isobutyl	2-OH	>100	94		2.50
1f	N	*tert*- butyl	2-OH	>100	>100		2.65
2	CH	H	3-OH	>100	51	>1000	2.96
2a	N	H	3-OH	100	50	>1000	1.01
2b	N	propyl	3-OH	>100	48		2.28
2c	N	isopropyl	3-OH	97	83	>1000	2.25
2d	N	butyl	3-OH	>100	67	>1000	2.73
2e	N	isobutyl	3-OH	>100	>100		2.50
2f	N	*tert*- butyl	3-OH	>100	>100		2.65
3	CH	H	4-OH	>100	>100	>1000	2.96
3a	N	H	4-OH	>100	97	>1000	1.01
3b	N	propyl	4-OH	>100	94		2.28
3c	N	isopropyl	4-OH	>100	78	>1000	2.25
3d	N	butyl	4-OH	>100	83	>1000	2.73
3e	N	isobutyl	4-OH	>100	>100		2.50
3f	N	*tert*- butyl	4-OH	>100	>100		2.65
4	CH	H	3-OCH_3_, 4-OH	>100	>100		2.89
4a	N	H	3-OCH_3_, 4-OH	>100	>100	186	0.94
4c	N	isopropyl	3-OCH_3_, 4-OH	>100	>100	39	2.18
4e	N	isobutyl	3-OCH_3_, 4-OH	>100	>100	46	2.43
4f	N	*tert*- butyl	3-OCH_3_, 4-OH	>100	36		2.58

^1^ The IC_50_ for vitamin C is 15 μM. ^2^ Values of molecular lipophilicity (clogP) were calculated by DataWarrior [17].

**Table 2 antioxidants-08-00090-t002:** Computed parameters (in kcal/mol) of O–H bond dissociation energy in the gas (g) and aqueous (aq) phases (BDE_g_, BDFE_aq_) and the electron donating capacity (ionization potential and electron transfer free energy) (IP_g_, ETFE_aq_) as well as acidity (pK_a_) of studied derivatives with unsubstituted pyrazine ring A (a) and corresponding chalcones, the guaiacyl derivatives 4c and 4e in addition to well-known antioxidants.

Compound	BDE_g_	IP_g_	BDFE_aq_	IFE_aq_	pK_a_	ETFE_aq_
1	84.9	188	85.7	107.8	10.9	77.5
1a	85.7	192.5	82.9	106.9	8.4	78.1
2	88.5	191.2	86.5	108.9	13.2	75.2
2a	88.5	192	87.5	108.7	13.4	75.9
3	85.2	184.6	83.9	103.5	11.1	77.5
3a	85.4	188.4	84.4	102.8	10.7	76.6
4	86	177.5	84.7	99.3	12.8	73.9
4a	85.8	178.4	84.8	99.2	12.0	75.2
4c	85.7	176.4	84.2	97.8	11.8	74.8
4e	85.9	176.4	86.4	99.2	13.3	74.9
Apigenin (4′-OH)	86.7	176.7	85.3	100.6	11.4	76.4
Quercetin (4′-OH)	79.3	176	80.2	98.8	8.9	74.8
Vitamin C	78.2	206.8	77.6	109	3.7	79.4

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
