# Peer review of "Antioxidant Activities of Alkyl Substituted Pyrazine Derivatives of Chalcones—In Vitro and In Silico Study"

_antioxidants, 2019, doi:10.3390/antiox8040090_

Round 1
Reviewer 1 Report
Dear author,
The manuscript “Antioxidant activities of alkyl substituted pyrazine derivatives of chalcones – in vitro and in silico study” has been reviewed. I carefully go through the data presented in tables and its explanation in the results and discussion section and found some discrepancies. Please address the enclosed comments. There are number of major and minor changes which authors should clarify and revise before publication of this article in Antioxidants.
Please check below.
Major comments
1. Authors should rewrite the “Abstract” (reduce the method part and increase the result part).
2. Please include NMR spectral data for synthesized compounds as a supplementary data.
3. Authors should mention about the effect of R1 moiety for radical scavenging (RS) activity to explain the different RS capacity between 4a and 4c/4e.
Minor comments
1. Line 84: Please define “DMSO”.
2. Table 1: There are two 3b in the list of compounds. Please check it.
3. Line 152 (Table 1): Please change “DPPH•a” to “DPPH•”.
Author Response
On behalf of all collaborators, I thank to the reviewer for the meaningful and well-meaning comments and suggestions.
Here are our answers to specific items:
Please check below.
Major comments
1. Authors should rewrite the “Abstract” (reduce the method part and increase the result part).
The Abstract has been rewritten in accordance to suggestion.
2. Please include NMR spectral data for synthesized compounds as a supplementary data.
We appreciate the comment/suggestion. However, since NMR data have already been published, the following text „The characteristics of all compounds (melting points, IR and NMR data) are available: for compounds 1, 2, 3 and 4 in Supplementary material of ref. [18], for 1a, 1b, 1d,1e, 1f, 3a, 3b, 3d, 3e, 3f, 4a, 4e and 4f in ref [6], 2a, 2b, 2d, 2e and 2f in ref. in [7] and 1c, 2c, 3c and 4c in ref. [8].“ is included in the section 2.1 at lines 86-90.
3. Authors should mention about the effect of R1 moiety for radical scavenging (RS) activity to explain the different RS capacity between 4a and 4c/4e.“ Minor comments
In accordance to the suggestions, at lines 274-276, the sentence „The alkyl isopropyl and isobutyl R1 substituents significantly increase molecular lipophilicty of 4c and 4e as compared with unsubstituted analogon 4a (Table 1).“ is added.
1. Line 84: Please define “DMSO”.
Done at line 90.
2. Table 1: There are two 3b in the list of compounds. Please check it.
Done.
3. Line 152 (Table 1): Please change “DPPH•a” to “DPPH•”.
Done.
Reviewer 2 Report
The manuscript shows data about the antioxidant activity of chalcone–like derivatives. In particular, they evaluated antioxidant activity in THP-1 cells against H2O2 by trypan blue exclusion assay. This in vitro experimental approach is not exhaustive to do an antioxidant profile. They should determine the redox status using specific fluorescent probe for the reactive oxygen species formation or others oxidative damage parameters.
Moreover, they found antioxidant effects for some chalcone–like derivatives after short (4 hours) and long (96 hours) treatment times suggesting their bifunctional antioxidant role. In particular, after a long treatment it is probably that chalcone–like derivatives exert their activity through the activation of Kelch-like ECH-associated protein 1 (Keap1)/Nuclear factor erythroid 2-related factor 2 (Nrf2) pathway. It is note that chalcones could activate the Keap1/Nrf2⁻ARE pathway through a Michael addition reaction with the cysteines of Keap1, which acts as a redox sensor and negative regulator of Nrf2. For the best chalcone–like derivatives the authors should evaluate their ability to active the Nrf2.
The lack of this knowledge has implication to make conclusion about the potential direct and indirect antioxidant activity of novel chalcone–like derivatives.
Author Response
On behalf of all collaborators, I thank to the reviewer for the meaningful and well-meaning comments and suggestions.
Here are our answers to specific items:
The manuscript shows data about the antioxidant activity of chalcone–like derivatives. In particular, they evaluated antioxidant activity in THP-1 cells against H2O2 by trypan blue exclusion assay. This in vitro experimental approach is not exhaustive to do an antioxidant profile. They should determine the redox status using specific fluorescent probe for the reactive oxygen species formation or others oxidative damage parameters.
We appreciate the comment of the reviewer and must agree with the statement that trypan blue exclusion assay is not exhaustive to do an antioxidant profile. However, the aim of this part of the study was primarily to see if the compounds could alter cytotoxic and growth suppressing effects of hydrogen peroxide, as was recently described for natural (aloe vera extract) #27 and synthetic antioxidants (DHP derivatives) #42 (added). There are good reasons why we have used the trypan blue exclusion assay. Namely, although it is not specific indicator or ROS damage, it is reliable indicator of the cytotoxicity of hydrogen peroxide. We used hydrogen peroxide not only because it can be cytotoxic, but also because it is the major (patho)physiological non-radical ROS and crucial mediator of redox signaling. The use fluorescent probes, like DCFH-DA, is also associated with disadvantages, such as scavenging by FCS and the lack of ROS-specificity, in particular in case of excess of hydrogen peroxide, as was done in the particular experiment. The alternative/complementary approach we prefer to use, i.e. the detection of the HNE-protein adducts, was not convenient either because of the non-adhesive growth of the THP-1 leukemic cells, which make them in particular convenient for the trypan blue exclusion assay. For similar reasons we also did not use MTT assay, which is problematic in case of hydrogen peroxide treatment. Finally, we like to stress the fact that many cells treated by pro-oxidants or ROS like hydrogen peroxide do tend to recover after initial damage, which we intended to study in these experiments to reveal of the tested compounds could affect the overall growth of the cells treated by hydrogen peroxide. Accordingly we revised the manuscript stating that more clearly in Discussion (marked yellow) and wishing to thank again for the constructive comment and hope the reviewer will find our answer and modification of the manuscript appropriate.
Moreover, they found antioxidant effects for some chalcone–like derivatives after short (4 hours) and long (96 hours) treatment times suggesting their bifunctional antioxidant role. In particular, after a long treatment it is probably that chalcone–like derivatives exert their activity through the activation of Kelch-like ECH-associated protein 1 (Keap1)/Nuclear factor erythroid 2-related factor 2 (Nrf2) pathway. It is note that chalcones could activate the Keap1/Nrf2⁻ARE pathway through a Michael addition reaction with the cysteines of Keap1, which acts as a redox sensor and negative regulator of Nrf2. For the best chalcone–like derivatives the authors should evaluate their ability to active the Nrf2. The lack of this knowledge has implication to make conclusion about the potential direct and indirect antioxidant activity of novel chalcone–like derivatives.
Again we are very thankful for very constructive suggestion, which is gladly accepted and commented within the text in lines 288-296. In addition, the proper references #44 and #45 have been added. This MOA may be in focus of further studies during which we will try to study in more details possible effects of the selected substances on redox signaling, lipid peroxidation and consequential protein modifications.
Reviewer 3 Report
The manuscript entitled "Antioxidant activities of alkyl substituted pyrazine derivatives of chalcones – in vitro and in silico study" is interesting in view of present pharmacological importance of chalcones. The paper described the alkyl substituted pyrazine derivatives and evaluated their antioxidant activity. It is carefully done and well written. The information generated are of major interest for the readers of Antioxidants. I recommend publication of this interesting paper in Antioxidants
Author Response
On behalf of all collaborators, I thank to the reviewer for reccomendation of our manuscript for publication in Antioxidants.
Round 2
Reviewer 2 Report
The authors answered in an exhaustive manner to both the questions about in vitro experimental approach adopted and the potential bifunctional antioxidant role of the chalcone-like compounds.
Minor points
They evaluated the cytotoxicity preliminary of the chalcone-like compounds by MTS after overnight treatment (about 18 hours) of HepG2 and THP-1. This short time of treatment allows only to evaluate the cytotoxicity in terms of cell viability, but not cell proliferation. So, they should correct the sentence “ … no considerable effects on cellular metabolism and proliferation of HepG2 and THP-1 cells” (page 4, line 142).
They wrote “significant cytotoxicity against THP-1 cells was detected only for the 2-OH derivate 1d (IC50 ≤ 30 μM)". At this regard, what does it mean the term “significant”? Does IC50 30 μM is different from IC50 36 μM (Table 1)? I think that from the biological point of view these values are very similar. They should review the sentence considering a group of relevant IC50;
They showed data from in silico methods in supplementary file. At this regard, to improve the understanding of the data, reference parameters standards should be added if appropriate. For example, does the score of 0.21 and 0.55 for Druglikeness and Bioavailability, respectively, is an evaluation great, good or sufficient?
Author Response
The authors answered in an exhaustive manner to both the questions about in vitro experimental approach adopted and the potential bifunctional antioxidant role of the chalcone-like compounds.
On behalf of all collaborators, I thank to the reviewer for comments and suggestions.
Here are our answers to the specific items:
Minor points
They evaluated the cytotoxicity preliminary of the chalcone-like compounds by MTS after overnight treatment (about 18 hours) of HepG2 and THP-1. This short time of treatment allows only to evaluate the cytotoxicity in terms of cell viability, but not cell proliferation. So, they should correct the sentence “ … no considerable effects on cellular metabolism and proliferation of HepG2 and THP-1 cells” (page 4, line 142).
Done in line 142.
They wrote “significant cytotoxicity against THP-1 cells was detected only for the 2-OH derivate 1d (IC50 ≤ 30 μM)". At this regard, what does it mean the term “significant”? Does IC50 30 μM is different from IC50 36 μM (Table 1)? I think that from the biological point of view these values are very similar. They should review the sentence considering a group of relevant IC50;
Corrected in lines 143-146.
They showed data from in silico methods in supplementary file. At this regard, to improve the understanding of the data, reference parameters standards should be added if appropriate. For example, does the score of 0.21 and 0.55 for Druglikeness and Bioavailability, respectively, is an evaluation great, good or sufficient?
Done and marked yellow and should stay yellow.
Kind regards
VS